# Emerging Tumor Biomarkers in Pancreatic Cancer and Their Clinical Implications

**DOI:** 10.3390/cimb47050347

**Published:** 2025-05-10

**Authors:** Dimitrios Stefanoudakis, Maximos Frountzas, Nikolaos V. Michalopoulos, Dimitrios Schizas, Dimitrios Theodorou, Konstantinos G. Toutouzas

**Affiliations:** 1First Propaedeutic Department of Surgery, Hippocration General Hospital, School of Medicine, National and Kapodistrian University of Athens, 11527 Athens, Greece; stefanoudak@med.uoa.gr (D.S.); froumax@med.uoa.gr (M.F.); nmichal@med.uoa.gr (N.V.M.); dimitheod@netscape.net (D.T.); 2First Department of Surgery, Laikon General Hospital, School of Medicine, National and Kapodistrian University of Athens, 11527 Athens, Greece; dschizas@med.uoa.gr

**Keywords:** cancer detection, cancer biomarkers, pancreatic cancer, microRNA, GATA6, L1CAM, MUC1

## Abstract

Pancreatic cancer is one of the deadliest malignancies, and this is attributed to the fact that it is diagnosed at a late stage and there are limited treatment options. Tumor biomarkers are used to improve early diagnosis, treatment, and decision-making and to estimate patients’ outcomes. This review aims to discuss the new functions of important biomarkers, such as miRNAs, GATA6, L1CAM, and MUC1 in pancreatic cancer. MiRNAs, including miR-21, miR-155, and miR-196a, are prognostic in PC and may be potential therapeutic targets through the regulation of oncogenic pathways and chemoresistance. GATA6, a transcription factor that controls tumor differentiation and immune escape, has been proposed as a pancreatic ductal adenocarcinoma (PDAC) subtyping marker and a predictor of chemotherapy response. L1CAM promotes tumor growth, invasion, and immune suppression, which leads to the formation of new metastases and perineural invasion. MUC1, a glycoprotein with altered glycosylation, is a marker of tumor progression, immune escape, and resistance to chemotherapy. These biomarkers can be combined into diagnostic panels that may increase the accuracy of the diagnosis and help to individualize the treatment plan. However, the present study is inconclusive, and more clinical evidence is needed to apply these biomarkers in clinical practice. More specific research should be directed towards the development of new targeted therapies that would act on these molecular targets and improve the prognosis and treatment of pancreatic cancer.

## 1. Introduction

In 2023, 1,958,310 cancer cases and 609,820 cancer-related deaths were reported in the USA. Pancreatic cancer (PaC) contributed to 64,050 cases and 50,550 deaths. PaC ranks as the leading cause of cancer related deaths among both men and women, with a rise in mortality rates observed in men [1]. The high mortality rate linked to PaC is mainly attributed to the challenges in detection methods, which lead to diagnosis in advanced stages, as well as the limited effectiveness of existing chemotherapy regimens. It is worth mentioning that almost 40% of patients diagnosed with PaC already have metastatic disease at the time of diagnosis and that the overall 5-year survival rate reaches 15% [2].

Under these circumstances, the need for a sensitive tumor biomarker which would lead to a timely and accurate diagnosis of pancreatic cancer becomes even greater (Table 1). Recent research has highlighted miRNAs that show levels of activity in pancreatic ductal adenocarcinoma (PDAC) [3]. For instance, combined plasma analyses for miR-21, miR-210, miR-155, and miR-196a discriminated patients with pancreatic cancer from normal healthy individuals with a sensitivity of 64% and a specificity of 89% [4]. These specific miRNAs could serve as both indicators for detection and as targets for treatment approaches. Current therapeutic methods involve the utilization of mimics to reinstate tumor-suppressing miRNAs or anti-miRs/antagomiRs to counteract oncomiRs. These strategies are designed to enhance effectiveness and reduce side effects compared to chemotherapy and radiation treatments [3].

MicroRNAs (miRNAs) are small, non-coding RNA molecules (approximately 18–25 nucleotides) that regulate gene expression post-transcriptionally by binding to complementary sequences in target messenger RNAs (mRNAs), leading to their degradation or translational repression. They play crucial roles in various biological processes, including development, differentiation, apoptosis, and tumorigenesis. The dysregulation of miRNAs has been implicated in numerous diseases, particularly cancer, where they can act as oncogenes or tumor suppressors [5,6].

miR-10b, miR-21, miR-23a, and miR-31 have important functions as key microRNAs in pancreatic ductal adenocarcinoma (PDAC) in terms of tumor growth, invasion, and resistance to therapy. miR-10b is a metastatic miRNA that increases the migratory and invasive properties of tumor cells by inducing EMT and its suppression has been found to hinder metastasis. [7,8,9]. miR-21 is one of the most highly expressed miRNAs in PDAC and acts as an oncomiR by repressing genes with tumor suppressor functions such as PTEN, PDCD4, and TPM1, thus enhancing tumor growth and gemcitabine resistance, and, therefore, is a potential diagnostic and therapeutic target [7,8,10]. miR-23a contributes to tumor progression through the suppression of tumor suppressor genes, including PTEN and E-cadherin, and also participates in the control of apoptosis and in the development of drug resistance, thereby reducing the sensitivity to chemotherapy [7,11,12]. The role of miR-31 is dual: it acts as an oncogene and a tumor suppressor according to the cellular context, and its expression is increased in PDAC, where it is associated with tumor growth, inflammation, and therapy resistance, including through the modulation of NF-κB signaling. Since there is a need for better biomarkers in pancreatic cancer, these miRNAs could be useful for enhancing tumor detection, predicting the prognosis, and designing individualized treatment plans, which require further clinical investigation [13,14,15].

**Table 1 cimb-47-00347-t001:** This table summarizes the key biomarkers in the main text, highlighting their types, functions, mechanisms of action, and clinical relevance, emphasizing in their roles in tumor progression, chemoresistance, immune evasion, and prognosis.

Biomarker	Type	Function/Mechanism	Diagnostic/Prognostic Value	References
MicroRNA	Non-coding RNA	Regulates gene expression and impacts cell growth, apoptosis, and chemoresistance	High expression levels of some miRNAs and/or low expression levels of others are linked with poor prognosis and chemoresistance	[16,17,18,19]
GATA6	Transcription Factor	Regulates differentiation and growth, and is involved in Wnt signaling pathway	Low GATA6 expression is associated with poor differentiation and survival	[20,21]
L1CAM	Cell Adhesion Molecule	Enhances cell migration, invasion, and immune evasion	High expression correlates with advanced stages and poor prognosis	[22,23,24]
MUC1	Glycoprotein	Aids in chemoresistance, immune evasion, and metastasis	Overexpression is linked to poor prognosis and treatment resistance	[25,26,27,28]

## 2. Emerging Tumor Markers in Pancreatic Cancer

### 2.1. MicroRNA Signatures

MicroRNAs, also known as miRNAs, are RNA molecules which play an important role in various biological processes by controlling gene expression. The process of their synthesis and function is depicted in Figure 1. Initially, miRNA genes are transcribed to form miRNA (miRNA), which then undergoes cleavage to produce precursor miRNA (pre-miRNA). In the cytoplasm, pre-miRNA is further cleaved to create a duplex. The mature miRNA regulates gene expression through interactions with mRNA, leading to either mRNA cleavage or the inhibition of translation depending on how the sequences of the mRNA match. Alterations in miRNAs have been associated with cancer during the last few years [16]. The effects of miRNAs are being enhanced by targeting genes simultaneously, which sometimes makes understanding gene behavior easier [29,30]. On the other hand, circulating miRNAs remain stable, either by binding to the chaperone protein Argonaute 2 (Ago2) or by being enclosed within vesicles, making them resistant to degradation by ribonucleases [31]. Argonaute 2 plays a pivotal role in RNA-induced silencing complex, facilitating gene regulation via RNA interference and microRNA-mediated silencing mechanisms. AGO2 possesses unique slicer activity, enabling the direct cleavage of target mRNAs. This slicer activity makes AGO2 pretty essential for processes like miRNA maturation and gene silencing near cells undergoing development or differentiation or even tumorigenesis [32,33]. Light is shed on gene regulatory networks by over 2500 identified microRNAs which target genes within molecular pathways [29,30].

MicroRNAs (miRNAs) are RNA molecules that do not code for proteins and control gene activity by attaching to the 3′ regions (3′UTR) of target mRNAs. This process can break down the mRNA and slow down its translation. In PDAC, miRNAs have significant roles in cancer progression. Some miRNAs, like miR-212, miR-196b, and miR-221-3p, help tumors grow by suppressing genes that control cell growth and survival mechanisms. On the other hand, miRNAs such as miR-451a, miR-142, and miR-519d-3p work as tumor suppressors by targeting pathways related to tumor development [17]. These small molecules could serve as potential targets for treating PDAC. Various approaches include using compounds that oppose miRNAs or imitate those that inhibit tumor growth to slow down cancer cell proliferation and invasion while enhancing responsiveness to chemotherapy [18]. An analysis of miRNA profiles in different tissues (Table 2) has revealed distinct differences between cancerous and healthy tissues [36,37]. Despite these breakthroughs, the early detection of cancer remains a challenge, underscoring the need for diagnostic biomarkers in pancreatic cancer research [7,38,39].

### 2.2. Role of MicroRNA in Diagnosis

Pancreatic cancer has been characterized by its difficulty in early diagnosis, as symptoms at the beginning are missing, resulting in a higher stage of disease and increased death rates. MicroRNAs seem to be a prognostic and potentially therapeutic option. Certain key miRNAs, such as miR-21, miR-155, miR-196a, and miR-210, are frequently found in high levels in pancreatic cancer patients. When combined with conventional pancreatic cancer biomarkers, like CA19-9, they show improved accuracy in predicting oncologic outcomes [7].

In a research study which examined 2555 serum miRNAs in pancreatic cancer and healthy individuals, 13 miRNAs that were significantly different between the two groups were demonstrated. These miRNAs also showed promising results in predicting the operability of pancreatic tumors, highlighting their importance in invasive diagnostics and treatment design [37]. A comprehensive meta-analysis showed that different molecular profiles of blood samples from pancreatic cancer patients were accurate for diagnosing cancer and miR-21 was the most reliable biomarker [10]. Another case–control study involving 409 pancreatic cancer patients identified 38 dysregulated miRNAs and two diagnostic panels based on them were developed. These panels demonstrated increased prognostic value, especially when used in combination with CA19-9, which significantly enhanced their diagnostic precision. This study underscores the potential use of such panels as noninvasive tools for detecting early-stage pancreatic cancer [68].

Another study examined the levels of miRNAs-21-23a, 100, 107, 181c, and 210 in both plasma and tissue samples from pancreatic cancer patients compared to a control group. The researchers discovered variations in miRNAs-21 and -210 in tissue samples, as well as miRNAs-181c and -210 in the plasma samples of pancreatic cancer patients when compared to controls. By combining tissue miRNAs-21 and -210 with increased serum CA19-9 levels, they achieved a 100% accuracy rate in diagnosing pancreatic cancer. Furthermore, plasma miR-181c exhibited a higher sensitivity and specificity than CA19-9 [69]. Utilizing data from the Gene Expression Omnibus (GEO) database, Shams et al. pinpointed 27 miRNAs with increased diagnostic ability for pancreatic cancer, particularly emphasizing the increased specificity and sensitivity of miR-1469 and miR-4530. Through multivariate Cox regression analyses, they developed models incorporating miR-125a-3p, miR-5100, and miR-642b-3p, with an AUC of 0.95, as well as a sensitivity of 0.98 and specificity of 0.97 for distinguishing pancreatic cancer patients from controls. Validation using clinical datasets further supported these identified miRNAs as noninvasive biomarkers for pancreatic cancer [70].

### 2.3. Role of MicroRNA in Prognosis

Certain small RNA molecules, like miR-21 and miR-221, might play a role in cell growth, death, and invasion. These molecules interact with the surrounding environment of tumors, affecting the action of chemotherapeutic agents. The activities of such RNAs are also affected by the gut bacteria composition. In addition, certain small RNAs like miR-21, miR-155, miR-196a, and miR-210 are overexpressed in pancreatic cancer tissues and could be used in distinguishing them from normal tissues, especially when they are evaluated in combination with conventional biomarkers, such as CA19-9. Higher levels of miR-21 are associated with greater survival and a favorable response to systematic treatment, whereas increased invasiveness and poorer prognosis are linked to miR-196a and miR-155, respectively. Some RNAs, like miR-34a or miR143/145, could suppress tumor growth as well [7].

Chemoresistance is a significant challenge in pancreatic cancer treatment. MiRNAs, like miR-483 3p and miR-21, have been linked to negative prognosis for pancreatic adenocarcinoma [36]. Moreover, upregulated miRNAs, like miR-21, miR-155, and miR-196a, have been associated with poor outcomes and increased resistance to chemotherapy, while tumor suppressor roles have been attributed to miR-34a and miR-145 [11]. MiRNAs, such as miR-21, miR-155, and miR-200, could modulate responses to chemotherapy. For instance, the overexpression of miR-21 is associated with a poor response to gemcitabine, which is a standard chemotherapeutic agent for pancreatic cancer. Nevertheless, restoration or inhibition of specific miRNAs could modulate chemosensitivity, offering potential therapeutic targets for pancreatic cancer [8].

MiR-34 is downregulated in pancreatic cancer, and its restoration impacts angiogenesis, apoptosis, cell cycle progression, and metastatic potential. Similarly, the miR-143/145 cluster’s downregulation has been correlated with disease progression, while its restoration inhibits cell proliferation. MiRNAs also have an impact on the metastatic potential of pancreatic cancer. In particular, miR-218 and miR-155 are linked with lymphatic metastasis, while miR-21, miR-10a, and miR-10b are associated with increased invasiveness and poor prognosis [8]. Finally, it has been noted that removing miR-34a leads to a faster progression of pancreatic cancer cells in pancreatic tissue, while advancing from early abnormal growths to invasive tumors has been noted within six months in a mouse experimental model. This rapid advancement is linked to an increase in inflammatory proteins, like TNF-α and IL-6, which create a hostile environment that supports the growth of such abnormal cells. Further analysis of mouse cells shows the activation of inflammatory and immunomodulatory pathways. These findings suggest that these molecular changes facilitate the development of malignant lesions in the pancreatic tissue [9].

### 2.4. Therapeutic Potential of MicroRNA and Its Limitations

MiRNAs, such as miR-143/145 and Mir-34a, have been found to be effective in slowing down tumor growth and making cancer cells more responsive to chemotherapy and radiotherapy. On the other hand, inhibitors which act against miR-21, miR-10a, and miR-212 have shown success in reducing tumor growth and preventing metastasis. The delivery of miRNAs has been improved by using vectors, nanoparticles, and exosomes, which provide precise distribution across cancerous tissues. For example, nanoparticles coated with oligonucleotide analogs have successfully delivered anti-miR-21 molecules to cancer cells, stopping tumor progression. Exosomes could also act as carriers which deliver miR-145-5p and significantly inhibit the proliferation and invasion of cancer cells [7].

Clinical trials exploring therapies based on miRNAs have yielded important outcomes. MRX34, a formulation containing miR-34a enclosed in liposomes, initially showed effectiveness against cancer cells, but it faced some challenges due to immune-related reactions. Another method, which was called siG12D LODER™, released siRNA targeting the KRAS(G12D) gene, leading to an improvement of the outcomes of chemotherapy [18]. Enhancing knowledge about miRNA-regulated pathways, like KRAS, AKT, JAK/STAT/WNT/β-catenin, and TGF-β, could lead to strategies for treating cancer. MiRNAs could play a role in regulating these pathways, as either suppressors or promoters of tumors, influencing cancer progression and resistance to chemotherapy [38].

Manipulating these pathways using miRNAs could offer some more therapeutic targets. For instance, reinstating tumor-suppressing miRNAs or blocking oncogenic miRNAs could hinder tumor growth and improve the efficacy of chemotherapy. Despite advancements in delivery systems, like vectors and nanoparticles, challenges still persist in ensuring accuracy and minimizing side effects [11]. Pancreatic cancer-related miRNAs can sometimes serve a dual role as both tumor suppressors and oncogenes (onco-miRs) impacting cancer progression and resistance to treatment. MiRNAs also play a role in autophagy, initially suppressing tumors but later aiding cancer survival under stress. Under these circumstances, inhibiting autophagy could offer potential therapeutic strategies for treating pancreatic cancer [12]. On the other hand, miRNAs regulate pathways in cancer development, such as MEK/ERK and PI3K/Akt, influencing cancer cell growth, metabolism, and metastasis, as Figure 2 and Figure 3 characteristically show. Therefore, increased levels of miRNAs in blood can act as markers for cancer [13].

In cases of cancer, miRNAs play a role in the progression of the disease and resistance to chemotherapy by controlling genes, like KRAS and TP53. Notably, miR-21, miR-155, and miR-221 are associated with cancer prognosis. These microRNAs might present diagnostic and prognostic value, since heightened levels of miRNAs in the bloodstream could reflect the cancerous load of patients. Furthermore, targeting modifications of miRNAs through DNA methylation and histone acetylation could represent another therapeutic approach [7]. One particular microRNA, MiR-1469-5p, shows an increase in both cancer tissues and blood samples, correlating with advanced stages of cancer and lymph node spread. It facilitates cancer cell growth and invasion by activating the NF-κB pathway, while directly targeting NDRG1, a suppressor of metastasis. Suppressing MiR-1469-5p leads to decreased NF-κB activity, increased NDRG1 levels, and the enhanced expression of E-cadherin, which hinders cancer cell invasion. This intricate relationship among MiR-1469-5p, NDRG1, the NF-kB pathway activation, and E-cadherin expression highlights its potential as both a biomarker for diagnostic purposes and as a potential target for therapeutic interventions in pancreatic cancer [15]. Dysregulated miRNAs, such as miR-1290 and miR-1246, serve as diagnostic markers, while miR-137 and miR-146a-5p enhance chemotherapy efficacy [14].

The miR-34 family—a crucial tumor-suppressing miRNA family—is oriented by genes on chromosomes 1 (miR-34a) and 11 (miR-34b/c). The tumor suppressor protein p53 plays a role in regulating miR-34 by interacting with its gene promoter. Alterations in miR-34 activity in cancer could arise from modifications, such as DNA methylation and histone deacetylation. Experimental treatments using miR-34 mimics, like MRX34, have demonstrated promising results by hindering tumor growth and boosting molecular sensitivity [77]. Furthermore, miR-34a targets key oncogenes, like c-MYC, CDK6, and c-MET, to promote cell cycle arrest and apoptosis. It downregulates anti-apoptotic proteins, such as Bcl-2, and inhibits EMT by targeting Snail1 and Notch pathways. MiR-34a biogenesis is also affected by p53 and could be modulated by epigenetic changes, like promoter methylation. Therapeutically, miR-34a mimics or nanotechnology-based delivery systems seem to be efficient in preclinical studies, inhibiting tumor growth and metastasis and enhancing chemosensitivity across various cancers. Additionally, miR-34a could modulate chemotherapy response and could be upregulated by dietary compounds, like curcumin and emodin, suggesting its potential utilization in complementary cancer therapies [78].

It has been discovered that miR-34a, miR-34b, and miR-34c show low expression levels in cancer cells MiaPaCa2 and BxPC3, which have elevated levels of target genes Bcl-2 and Notch1/2. When miR-34 was restored, it led to the downregulation of these target genes, which hindered cancer cell growth and triggered apoptosis as well as cell cycle arrest, making these cells more sensitive to chemotherapy and radiation treatments. In cancer stem cells (CSCs) marked as CD44+/CD133+, restoring miR-34 reduced the CSC population by 87% by suppressing their growth and decreased tumor formation by lowering Bcl-2 and Notch1/2 levels. These results indicate that using miR-34 mimics could offer a treatment approach for p53-related pancreatic cancer by targeting both cancer cells and CSCs [79]. Another study conducted an analysis of exomiR-34a derived from exosomes obtained from HEK293 cells, which demonstrated its delivery into pancreatic cancer cells. ExomiR-34a notably hindered the proliferation of Panc28 and MiaPaCa-2 cancer cells, while causing harm to pancreatic cells. It boosted miR-34a levels, reduced Bcl-2 expression, and triggered apoptosis by enhancing the presence of apoptotic proteins Bax and P53. Moreover, treatment with exomiR-34a decreased tumor growth in a xenograft mouse model by reducing Ki67 and Bcl-2 levels and increased apoptosis within tumor tissues [80].

The expression of miR-34a is notably lower in cancer cells than in pancreatic epithelial cells. When miR-34a is overexpressed, it hinders the migration, invasion, and growth of cancer cells, while triggering apoptosis. MiR-34a directly acts on Snail1 and Notch1, reducing their levels and thereby impeding the epithelial–mesenchymal transition (EMT). When Snail1 is overexpressed, it counters the impact of miR-34a on migration and invasion, emphasizing Snail1’s significance in EMT control. Similarly, Notch1 plays a role in the interaction between miR-34a and cell growth as well as apoptosis. In an animal study, mimicking miR-34a significantly reduced the tumor size in mice, confirming what was observed in lab experiments [81]. In another study which included 85 cancer patients treated with the miR-34a mimic MRX34, pharmacokinetic and pharmacodynamic analyses confirmed dose-dependent miR-34a delivery and target gene modulation. The efficacy results showed partial responses in three patients and stable disease in sixteen patients, with a median duration of 19 weeks [82].

MiR-34 and miR-200 are controlled by p53, while miR-17-5p and miR-20a are activated by MYC. Ongoing clinical trials are investigating the potential of using miRNAs as biomarkers for cancer detection and therapy prognosis, while exploring them as targets for treatment. Noteworthy therapies based on miRNAs include MRX34 (mimicking miR-34a), TargomiR (mimicking miR-16), MRG 106 (inhibiting miR-155), INT 1B3 (mimicking miR-a-193a-3p), and TTX MC138 (inhibiting miR-10b). These therapies have shown varying degrees of success, underscoring the importance of enhancing delivery systems and safety protocols [83]. The use of nanotechnology has emerged as a way to deliver miR-34a effectively. Different types of systems, such as lipid-based nanoparticles (LNPs), polymeric nanoparticles (PNPs), mesoporous silica nanoparticles (MSNs), gold nanoparticles (GNPs), and viral vectors, have been developed to improve the stability of the targeting ability and availability of miR-34a. Despite facing challenges such as immune activation against the host, these innovative delivery techniques offer potential for enhancing cancer treatments based on miR-34a [84,85].

### 2.5. GATA Family

GATA6, a member of the GATA family found on chromosome 18, is prominently active during the stages of embryogenesis in the endoderm and mesoderm. It plays a crucial role in shaping organs like the heart, lungs, and digestive system. Alterations in GATA6 levels have been linked to cancer. In pancreatic carcinoma, GATA6 contributes to tumor growth by influencing the Wnt signaling pathway [20]. Further experimental exploration based on the deactivation of Gata4 and Gata6 in the pancreas demonstrated that simultaneous absence led to pancreatic agenesis characterized by disruptions in progenitor cell growth and branching morphogenesis. Some targets affected by Gata factors are Pdx1, Neurog3, and Cpa1, which are essential for tissue formation and differentiation [86].

It has been shown that GATA4 is found in cancer tissues, and it is correlated with the level of tumor differentiation, indicating a role in cancer progression. When GATA4 was injected into cancer cells where it would not normally be present, it resulted in decreased cell growth and colony formation. In mouse models, an increased presence of GATA4 slowed down tumor growth, indicating its ability to suppress tumors. Analysis of gene expression revealed that elevated levels of GATA4 influenced the genes associated with cell specialization and activated crucial pathways, like MAPK and JAK/STAT, necessary for cell growth and differentiation. Additionally, reintroducing GATA4 activated the tumor suppressor gene P53 [87].

### 2.6. Diagnostic Value of GATA6

Pancreatic adenocarcinoma is characterized by high activity levels of GATA3 and GATA6 while GATA2 and GATA4 exhibit low activity compared to normal pancreatic tissue. The expression of GATA4 and GATA6 are linked to the differentiation level and grading of pancreatic cancer, whereas lower GATA4 and higher GATA6 levels are connected to survival rates. Despite the absence of any direct association with survival outcomes, GATA3 is related to cell infiltrative ability, especially for CD8+ T cells, dendritic cells, and neutrophils. On the other hand, analysis at a single cell level revealed expression patterns for members of the GATA family within the tumor microenvironment. In particular, exhausted CD8+ T cells express high levels of GAT3, whereas acinar and malignant cells show a significant presence of GAT4, while ductal epithelial as well as malignant cells exhibit notable amounts of GAT6 [21].

Researchers have shown that Wnt signaling plays an important role in maintaining a balance between stem cell upkeep and differentiation in both healthy tissues and cancer. In cases of pancreatic carcinoma associated with RNF43 mutations, inhibiting Wnt has led to tumor differentiation, while cell death is diminished, though resistance could arise. Screening using the CRISPR system has revealed that the absence of the histone acetyltransferase p300 (EP300 gene) causes resistance to Wnt inhibitors [88].

In mice lacking Gata6 gene expression, there was an increase in acinar to ductal metaplasia (ADM) and faster tumor growth driven by KRAS-G12V mutation. After 5 weeks, these mice showed ADM and inflammation progress, leading to neoplasia (PanINs) with a higher occurrence of carcinomas by 25–28 weeks compared to control mice. All mice that lacked Gata6 elements developed large cancerous tissues, some of which spread to the liver and abdomen over a period of 50–70 weeks. The absence of Gata6 also led to increased EGFR levels, indicating that the Gata6 status could impact responses to EGFR inhibitors [89]. In mice with KRAS mutations, nicotine sped up the development of cancer by promoting cell changes and increasing the occurrence of malignant pancreatic lesions. Nicotine triggered a transformation in acinar cells by reducing GATA6 and Mist1 levels through the signaling pathways, resulting in the higher activity of progenitor cells and the enhanced expression of genes associated with stemness like Sox9 [90].

### 2.7. GATA6 and Its Contribution in Prognosis

High GATA6 expression correlates with better overall survival (OS) and disease-free survival (DFS), while low GATA6 expression is linked to larger tumors, a higher percentage of lymph node metastasis, and poorer differentiation. Among GATA6-high patients, elevated CK5 expression indicates worse overall survival (OS) and disease-free survival (DFS) compared to those with low CK5 expression. Conversely, in the GATA6-low group, low CK5 expression is associated with worse OS [91].

A study involving 130 patients with PDAC found that 44% were GATA6-low and 56% GATA6-high based on immunohistochemistry (IHC) expression. GATA6-low tumors had higher progression rates and poorer overall response rates to FOLFIRINOX (mFFX) chemotherapy compared to GATA6-high tumors, which had better OS rates, especially in mFFX-treated patients [92]. Another study found that the loss of GATA6 was essential but not sufficient alone for the expression of ΔNp63 and the basal phenotype in PDAC. It required the concurrent loss of HNF1A and HNF4A, likely through epigenetic silencing. GATA6 regulated tumor cell plasticity and immune evasion and its loss induced a basal transcriptional program driven by ΔNp63. Overexpressing GATA6 reduced ΔNp63 and basal marker expression, while GATA6 knockdown increased them. HNF1A and HNF4A were vital for maintaining the classical PDAC phenotype, and their loss combined with GATA6 promoted the basal phenotype. GATA6 loss was also correlated with immune modulation features, including reduced MHC class I gene expression and decreased CD8+ T cell infiltration rates, leading to more aggressive PDAC behavior and increased metastasis [93]. Finally, there was no connection discovered between SMAD4 mutations and GATA6 expression, suggesting that GATA6’s predictive ability was not affected by the status of SMAD4 [94].

In another study, it was highlighted that GATA6 was expressed in 81% of PDAC tissues, with consistent expression in normal tissue and PanIN, but reduced expression in lymph node metastases. GATA4 was expressed in 42% of tumors and decreased during tumor progression [95]. Non-smokers had a higher percentage of GATA4-positive tumors. Early-stage and well-differentiated tumors consistently expressed GATA6, but GATA6-negative tumors were associated with perineural invasion. Both GATA4 and GATA6 expressions correlated with lower Ca 19.9 levels and improved DFS, with a trend towards better OS [95]. Additionally, in PDAC patients from two randomized phase III adjuvant chemotherapy trials, ESPAC-3 and ESPAC-4, low GATA6 expression was significantly associated with a shorter OS, with a median OS of 26.1 months compared to 42.4 months for high expression [96]. Decreased GATA6 expression was associated with changes in cell differentiation and a basal-like molecular profile, leading to reduced survival rates and poor reactions to 5 fluorouracil (5 FU)/leucovorin treatment [97]. It was observed that GATA6 hindered EMT and decreased cell spreading, indicating a role in suppressing tumor growth. The absence of GATA6 led to changes in tumor characteristics and a shift towards a basal phenotype, which was associated with patient outcomes and decreased responses to adjuvant chemotherapy, like 5 fluorouracil/leucovorin. Experiments involving silencing and overexpressing GATA6 in PDAC cell lines along with analyzing GATA6 ChIPSeq and RNA Seq data supported its function in maintaining traits [98]. Lastly, the COMPASS trial showed that low or lacking GATA6 expression indicated the basal subtype, which was associated with worse prognosis and resistance to chemotherapy. Therefore, the utilization of GATA6 as a biomarker for subtyping PDAC was implicated [99].

### 2.8. Therapeutic Potential of GATA6

EZH2 (Enhancer of Zeste Homolog 2) promoted PDAC progression and de-differentiation by increasing tumor cell invasion and suppressing differentiation pathways [100]. EZH2-deficient mice showed reduced PDAC incidence and liver metastases. EZH2 downregulated the transcription factor GATA6, which is essential for maintaining epithelial differentiation and suppressing pancreatic carcinogenesis, by direct transcriptional repression through H3K27me3 at the GATA6 promoter. Restoring GATA6 expression in EZH2-depleted cells shifted PDAC towards a more differentiated and less aggressive phenotype. Targeting EZH2, genetically or pharmacologically, increased GATA6 expression and reduced tumor aggressiveness, suggesting that EZH2 inhibitors could be a therapeutic strategy to induce a less aggressive, more therapy-susceptible PDAC subtype [101]. Another study demonstrated that the loss of HNF4A and GATA6 in PDAC induced a metabolic switch to a squamous-associated profile, marked by increased glycolysis regulated by glycogen synthase kinase 3 beta (GSK3b). The inhibition of GSK3b selectively impeded glycolysis and cell proliferation in squamous PDAC cell lines (PDCLs), but a subset quickly developed drug tolerance. An assay for transposase-accessible chromatin sequencing (ATAC-seq) revealed that chromatin accessibility patterns could predict responses to GSK3b inhibitors, with drug-tolerant cells exhibiting access to an amplified WNT gene program. These cells produced WNT ligands to activate WNT signaling, enabling survival despite GSK3b inhibition [102].

P300 was crucial for maintaining sensitivity to Wnt inhibitors by controlling GATA6, a significant transcription factor. When p300 was lost, GATA6 levels dropped, triggering a shift from the PDAC subtype to an aggressive basal-like/squamous subtype, increasing tumor aggressiveness and reducing differentiation. In PDAC cell lines resistant to RNF43 mutations, mutations which lead to GATA6 loss seemed to diminish the effectiveness of Wnt inhibitors. Restoring GATA6 expression in p300 cells brought back sensitivity to Wnt inhibitors, underlining the significance of the p300/GATA6 interplay in drug responses [88].

Moreover, GATA6 impaired the stem-like properties of PDAC cells by reducing cell proliferation, colony formation, and resistance to gemcitabine. The overexpression of GATA6 decreased the expression of stemness-related markers and modulated the Wnt/β-catenin signaling pathway by reducing β-catenin expression and nuclear translocation, thus inhibiting Wnt/β-catenin target genes. Conversely, GATA6 silencing increased β-catenin levels and activity. ChIP and dual-luciferase assays confirmed GATA6’s direct binding to the β-catenin promoter, suppressing its transcription [103]. UTX, a lysine-specific demethylase (KDM6A), was significantly downregulated in PDAC tissues compared to normal pancreatic tissues, correlating with poor differentiation and prognosis. Functional assays revealed that UTX suppressed PDAC cell growth, migration, and invasion, and GATA6 was identified as a key activator of UTX transcription, with its loss leading to reduced UTX expression and contributing to PDAC progression [104]. Disabling GATA6 led to a decrease in the formation of cell colonies, indicating a disadvantage in growth. Experiments with xenograft models further confirmed the importance of GATA6 in tumor development. Deactivating GATA6 triggered an interferon response and increased cell death [105].

Zhou et al. found that the expression of GATA6 AS1 was reduced in PDAC tissues when exposed to oxygen conditions, which was linked to unfavorable clinical features in PDAC patients. Experimental tests demonstrated that GATA6 AS1 hindered the growth, invasion, movement, and transition of PDAC cells both in vitro and in vivo. More specifically, GATA6 AS1 lowered SNAI1 mRNA stability by blocking the mass and obesity associated protein (FTO) in a dependent manner, resulting in the removal of m6A from SNAI1 mRNA. Hypoxia induced the E26 transformation-specific sequence 1 (ETS1), which was identified as a regulator that decreased the transcription of GATA6 AS1, thus boosting SNAI1 mRNA stability and expression. The GATA6 AS1/FTO/SNAI1 pathway seemed to play a role in this advancement. The spread of PDAC under low oxygen conditions suggested that targeting this pathway could be a promising approach for treating hypoxic pancreatic cancers [106].

### 2.9. L1CAM (L1 Cell Adhesion Molecule)

L1CAM is highly expressed in PDAC. Schwann cells release L1CAM, which attracts PDAC cells through MAP kinase signaling. Moreover, L1CAM increases the levels of metalloproteinases MMP-2 and MMP-9 in PDAC cells by activating STAT3, thus facilitating invasion. Administering anti-L1CAM antibodies notably decreased nerve invasion in a mouse model [22,107]. Interestingly, high L1CAM levels have been found in PDAC cells. A study revealed that exposure to myofibroblasts (PMFs) or treatment with transforming growth factor beta 1 (TGF-β1) triggered L1CAM production in the human pancreatic duct cell line H6c7. This process did not rely on Smad proteins, but necessitated JNK activation resulting in increased Slug, a transcription factor binding to the L1CAM promoter. The presence of L1CAM led to H6c7 cells developing resistance to chemotherapy and becoming more migratory. This same pattern was observed in TGF-β1 PDAC cell lines Colo357 and Panc1, suggesting that PMFs played a role in the cancerous transformation of pancreatic ductal cells by stimulating L1CAM expression through TGF-β1 and JNK/Slug pathways [108]. During PDAC progression, L1CAM expression increased in the epithelium, correlated with a less favorable prognosis. L1CAM facilitated the movement and accumulation of immune-suppressing T cells (T-regs) and a unique subset of CD4+CD25 CD69+ T cells within the tumor’s microenvironment. These T-regs hindered the activity of effector T cells (T-effs), fostering an immune-suppressive setting that aided tumor growth and evasion from responses. L1CAM expression boosted PDAC cell tumorigenicity, invasiveness, and resistance to cell death [109].

### 2.10. Diagnostic and Prognostic Value of L1CAM

It has been demonstrated that when L1CAM is expressed in epithelial cells, it boosts the movement and gathering of T-regs, while diminishing the growth of T-effs and transforming their characteristics to an immune-suppressive state. This mechanism is facilitated by the release of substances on L1CAM, primarily TGF-β1. Moreover, under the influence of L1CAM, CD69 levels are increased and CD25 levels are decreased, hindering the replication of autologous T-effs [23]. L1CAM is expressed in tumors, promoting cancer aggressiveness and resistance to treatment and serving as an indicator of poor prognosis. It contributes to CSC features, like self-regeneration DNA repair processes, mechanisms that expel drugs, dormancy, and the evasion of the tumor. The participation of L1CAM in the transition from EMT and resistance to chemotherapy further emphasizes its importance in cancer advancement [24].

In a study that analyzed 107 surgically resected PDAC specimens, it was found that L1CAM was positively expressed in 23 cases (21.5%), primarily at the tumor’s invasive surface. This expression was significantly correlated with advanced histological grades, lymph node involvement, and distant metastasis. Patients with positive L1CAM expression had significantly shorter OS, both in univariate and multivariate analyses, indicating that L1CAM is an independent prognostic factor for poor oncologic outcomes [110].

A systematic review of 45 studies highlighted how full-length L1CAM (L1CAM-FL) and its cleaved forms could be identified in various cellular environments and influence cancer cell behavior through different signaling pathways, such as integrins, FAK/Src, PI3K/Akt, FGFR, Ezrin, ERK, NF-κB, and L1CAM-FL. Proteolytic cleavage and exosome formation resulted in soluble L1CAM forms that retained functional similarities to the full-length protein, but also introduced unique mechanisms, impacting tumor progression indirectly [111]. It was found that L1CAM-FL interacted with integrins, leading to the constitutive activation of the NF-κB pathway. This activation was mediated through the upregulation of IL-1β expression. The study showed that this interaction and subsequent signaling were crucial for cell proliferation and tumor growth. Knockdown experiments of L1CAM or integrins (especially α5-integrin) resulted in reduced IL-1β levels and NF-κB activity, highlighting their role in maintaining the malignant properties of PDAC cells. Moreover, inhibiting the integrin-binding site on L1CAM (mutating RGD to RGE) or using neutralizing antibodies significantly impaired NF-κB activation and tumor growth. These findings suggested that targeting the L1CAM–integrin interaction could be a potential therapeutic strategy for PDAC [112]. Eventually, the potential use of L1CAM as a biomarker for cancer diagnosis and prognosis was found to be prominent, since L1CAM identifies cells that initiate metastasis with stem cell-like features and resistant traits towards chemotherapy [113,114].

### 2.11. L1CAM’s Therapeutic Potential

Therapeutic strategies aimed at targeting L1CAM include monoclonal antibodies, radioimmunoconjugates, and CAR-T cells, showing promising results in experimental models. These methods are designed to either counteract L1CAM’s function or transport substances to tumors expressing L1CAM, with the goal of increasing treatment effectiveness and overcoming resistance to chemotherapy [24]. L1CAM enhances cell proliferation and tumor growth by engaging integrin signaling and activating NF-κB. Kiefel et al. generated PT45-P1 cell lines expressing various L1CAM domains and found that L1CAM-FL supported cell proliferation and tumor growth in vivo. L1CAM-FL expression increased IL-1β and NF-κB activity, which are crucial for these processes, through integrin-mediated signaling pathways involving integrin-linked kinase (ILK). Importantly, disrupting the integrin-binding motif of L1CAM impaired its ability to promote these effects [112]. Moreover, L1CAM was found to be expressed in cancer tissues and cell lines, which are associated with aggressive tumor characteristics and lower overall survival rates. Increased levels of L1CAM facilitated cell growth, movement, invasion, tumor development, and the spread of cancer, through the activation of the PI3K/Akt signaling pathway. High levels of L1CAM were connected to decreased sensitivity to the chemotherapy drug oxaliplatin [115]. The research delves into how silencing L1CAM impacts cancer cells, specifically focusing on the Capan-2 cell line. Qiwen et al. used lentivirus-mediated hairpin RNA (shRNA) to reduce L1CAM expression, leading to a decrease in cell growth and invasion and an increase in the number of cells in the G0/G1 phase of the cell cycle, indicating a halt in cell division. However, silencing L1CAM did not notably trigger cell death. The decrease in L1CAM also triggered the activation of the p38/ERK1/2 signaling pathway [116]. By creating cell lines resistant to 5-FU from the Panc-03.27 cancer cell line, the investigators noted that these resistant cells displayed traits associated with EMT, such as increased invasiveness and the high expression of mesenchymal markers. Analysis using microarrays indicated an upregulation of the L1CAM pathway in these cells to chemotherapy, where L1CAM expression was found to be regulated by the transcription factor Slug rather than β-catenin. Experiments testing functionality showed that L1CAM played an important role in facilitating the invasive potential of these cells and its suppression led to reduced invasiveness and proliferation [117].

Using immunohistochemical assays, another study discovered that although CD3+ T cells and SMA+ fibroblasts were evenly spread in chronic pancreatitis (CP) and PDAC, CD4+ and CD8+ T cells were notably lower in number, while CD25+(CD4+) as well as FoxP3+(CD4+) regulatory T cells were elevated in PDAC. Macrophages were more abundant in CP [118]. In addition, in PDAC, they were positioned closer to carcinoma cells alongside CD T cells. The levels of FoxP3 and L1CAM rose from CP to PDAC, whereas vimentin expression remained high in both cases. Differentiated tumors and CPs exhibited similarities, whereas moderately to poorly differentiated tumors displayed more pronounced differences. The above research study pinpointed connections between elements with epithelial/carcinoma cells, like the correlation between CD4+ and FoxP3+CD4+ T cell presence with FoxP3 expression within PDAC cells, as well as the association of SMA+ fibroblasts with L1CAM expression and proliferation within PDAC cells. There were increased levels of FoxP3, vimentin, and L1CAM in PDAC cells, along with tumor-associated macrophage localization linked to tumor grades. Based on the results of the multivariate survival analysis, it was found that undergoing surgery at an early age had an impact on the prognosis of patients with PDAC. This indicates that the presence of stroma may play a role in inducing changes even in precursor cells during chronic pancreatitis [118].

Recent research delves into how immune cells like Tregs, TAMs, MDSCs, and cancer stem cells contribute to suppressing the immune response in PDAC. In PDAC, the TME creates an environment that hinders tumor immune reactions by utilizing various immune cells and cytokines that promote tumor development and spread. Particularly, Tregs and TAMs are highlighted for their roles in promoting immunosuppression-aiding tumor cells in evading detection by the system. There seems to be the potential for immunotherapy treatments like checkpoint inhibitors and combination therapies to counteract these mechanisms [119].

### 2.12. MUC1 (Mucin 1)

Mucins are glycoproteins which line epithelial cells and are involved in the lubrication and protection of the gut. In PDAC, mucins contribute to cancer progression through various mechanisms, including promoting tumorigenesis, enhancing metastasis, and inducing chemoresistance. Key mucins like MUC1, MUC4, MUC5AC, and MUC16 play significant roles in these processes. MUC1, for instance, is highly expressed in PDAC and is associated with poor prognosis and chemoresistance. MUC4 is linked with aggressive tumor behavior and poor survival outcomes. MUC5AC is considered to be a marker for early pancreatic neoplasms and is associated with cancer progression. MUC16 is involved in tumor metastasis and is combined with CA19-9 for prognostic evaluation [25].

MUC1, a highly glycosylated transmembrane protein, is overexpressed and aberrantly glycosylated in many cancers, including PDAC. This overexpression is linked to various mechanisms that confer resistance to chemotherapy, such as interference with apoptosis, reduction in drug uptake, increased drug efflux, and metabolic reprogramming. Specifically, MUC1 promotes the expression of multidrug resistance (MDR) genes, including ABCC1, which encodes the MRP1 protein, thereby facilitating the exodus of chemotherapeutic drugs from cancer cells. Additionally, MUC1 enhances the metabolic pathways that support cancer cell survival under chemotherapeutic stress [120]. Furthermore, MUC1 is a high-molecular-weight transmembrane glycoprotein that normally provides lubrication and protection to epithelial cells. However, its aberrant overexpression is linked to cancer progression, invasion, and metastasis. The findings indicate that MUC1 promotes tumor growth by affecting multiple signaling pathways such as PI3K/AKT, MEK/ERK, and WNT/β-catenin, among others. MUC1 levels can serve as a diagnostic marker and prognostic indicator, with high expression levels correlating with poor prognosis [121].

### 2.13. MUC1 as a Potential Diagnostic Marker

The significance of Muc1, Muc4, and Muc5AC has been highlighted in the progression of cancer using a KrasG12D;Pdx1-Cre murine model. Rachagani et al. noted a rise in the expression of these mucins from PanIN to advanced PDAC, which was correlated with an increase in inflammatory cytokines and the infiltration of macrophages [26]. Further research delved into the roles played by MUC1 in advancing cancer. MUC1 is excessively expressed in different types of epithelial cancers, exhibiting notable differences from its normal form in terms of structure, cellular location, and function. Within cancer cells, MUC1 impacts signaling pathways and controls gene expression both during transcription and after transcription. In cancer progression, MUC1 contributes to facilitating tumor growth, invasion, and metastasis, as it provides resistance against apoptosis and chemotherapy. Additionally, the influence of MUC1 on shaping the environment and its probable involvement in maintaining cancer stem cells highlights its significance in understanding cancer biology [27]. In cells, MUC1 acts as a shield, moisturizer, and lubricant that shields epithelial cells from harm and infections. However, in cancer cells, abnormal glycosylation and the increased expression of MUC1 often leads to processes like tumor invasion, generalized spread to parts of the body, the formation of blood vessels, and apoptosis. The clinical implications of MUC1 are significant, as it serves as a tool for the detection and prognosis of cancers, particularly epithelial adenocarcinomas found in organs, such as the lungs, liver, colon, breast, pancreas, and ovaries [28]. Qu et al. investigated the role of MUC1 expression in pancreatic cancer and its potential for targeted therapy using the 213Bi-C595 radioimmunoconjugate. MUC1 was overexpressed in approximately 90% of pancreatic cancer samples and cell lines. The 213Bi-C595 conjugate demonstrated specific cytotoxicity to MUC1-positive pancreatic cancer cells in a concentration-dependent manner, inducing apoptosis and reducing cell viability significantly compared to controls. These results suggest that MUC1 is a viable target for treating pancreatic cancer, particularly for micro-metastases or minimal residual disease, using 213Bi-C595 radioimmunoconjugate therapy [122].

The development of a new nanomaterial-based electrochemical biosensor for the detection of the MUC1 biomarker seems to be significant for early diagnosis, monitoring tumor progression, and cancer treatment. Using a mini-emulsion polymerization method, methacrylate-based nano-polymers were synthesized and functionalized with IMEO and Concanavalin A lectin to enhance specificity for MUC1 detection. The biosensor demonstrated excellent performance in terms of sensitivity, reliability, and its rapid response time, with a linear detection range of 0.1–100 U/mL and a response time of 20 min. Validation studies showed that the biosensor could accurately detect MUC1 levels in real blood serum samples and it was comparable to commercially available ELISA kits. The developed nano-polymeric biosensor offered a promising low-cost, efficient, and portable solution for cancer diagnosis and monitoring [123]. Another study investigated the detection of the MUC-1 biomarker, which is overexpressed in various malignant epithelial cells, using a novel aptamer-based Immuno-Loop-Mediated Isothermal Amplification (Im-LAMP) technique. This method leverages the high affinity of MUC-1 aptamers for MUC-1, enabling ultrasensitive and specific detection. The cycle time of this technique is linearly dependent on the logarithm of the MUC-1 concentration, achieving a detection limit as low as 120 molecules. Im-LAMP demonstrates superior sensitivity and specificity compared to other methods, and has been successfully applied in human blood serum analysis [124]. Finally, research introduced an innovative sensor based on a nanocomposite of silver-doped copper oxide integrated with polyaniline (Ag-CuO@PANI). This composite was synthesized using a hydrothermal method and then deposited on a fluorine-doped tin oxide (FTO) electrode, followed by the conjugation of an anti-MUC1 antibody via a cysteamine linker. The resulting immunosensor was characterized using various techniques, confirming its structural and electrochemical properties. The sensor demonstrated excellent analytical performance, with a low detection limit of 3.23 pg/mL, high sensitivity, and a wide linear detection range for MUC1 concentrations [125].

### 2.14. Prognostic and Therapeutic Significance of MUC1

It has been observed that mucins present changes in their expression levels and sugar structures as epithelial malignancies progress. Combining mucin-based biomarker panels with markers and advanced technologies, such as biopsies and imaging, is suggested to be an effective method for early cancer detection that can lead to improved clinical care and patient survival [126]. A research team reviewed 34 studies involving 3900 patients. The outcomes revealed that MUC1 had a sensitivity of 0.84 and specificity of 0.60 with an SROC curve area of 0.8875, indicating its promising capabilities. Meanwhile, MUC4 demonstrated sensitivity (0.86) and specificity (0.88), whereas MUC5AC showed sensitivity (0.71) and specificity (0.60). Notably, MUC16 displayed accuracy with an SROC area of 0.9185. These results suggest that mucins like MUC4 and MUC16 could be indicators for detecting PC [127]. In another study, MUC1 expression was studied in 158 patients using immunohistochemistry on tissue microarrays. The findings revealed that high cytoplasmic MUC1 expression was linked to DFS and OS. Notably, patients with low MUC1 expression showed outcomes regardless of whether they received gemcitabine treatment or underwent surveillance without it [128].

Established stable cell lines with reduced MUC1 expression have shown that inhibiting MUC1 significantly decreased cell proliferation, migration, invasion, and survival, while increasing apoptosis. These effects were mediated through the p42–44 MAPK, Akt, Bcl-2, and MMP13 pathways. In vitro experiments revealed that MUC1 knockdown made cells more sensitive to the chemotherapeutic drugs gemcitabine and 5-fluorouracil. In vivo, MUC1 knockdown cells showed reduced tumor growth in SCID mice [129]. In addition, PDAC cells with high MUC1 expression exhibited increased resistance to chemotherapeutic drugs, such as gemcitabine and etoposide, compared to cells with low MUC1 expression. This resistance was linked to the upregulation of multidrug resistance (MDR) genes, particularly ABCC1, ABCC3, ABCC5, and ABCB1. The study highlighted that MUC1 upregulated the MRP1 protein encoded by ABCC1 via an Akt-dependent pathway in some cells, while in others, it involved an Akt-independent mechanism. The cytoplasmic tail of MUC1 was shown to be directly associated with the promoter region of the ABCC1 gene, suggesting a role as a transcriptional regulator. Knockdown of MUC1 resulted in decreased drug resistance, underscoring the potential of targeting MUC1 to enhance the efficacy of chemotherapy in PC patients [130].

In addition, researchers found that MUC1 expression was significantly induced in cells that had acquired resistance to chemotherapy. This induction occurred at both transcriptional and post-translational levels, enhancing drug resistance through the upregulation of the ATP-binding cassette transporter B1 (ABCB1). Jin et al. demonstrated that MUC1 stimulated EGFR activation and nuclear translocation, which in turn increased ABCB1 expression. Targeted inhibition of EGFR or ABCB1 using shRNAs and pharmacological inhibitors effectively reversed chemoresistance. The co-administration of inhibitors targeting the MUC1-EGFR-ABCB1 axis with paclitaxel significantly inhibited tumor growth and relapse in a xenograft mouse model [131].

Wu et al. developed a humanized MUC1 antibody (HzMUC1) targeting the interaction region between MUC1-N and MUC1-C. This antibody was conjugated with monomethyl auristatin E (MMAE) to create HzMUC1-MMAE. The results demonstrated that HzMUC1-MMAE effectively bound to MUC1 on the surface of pancreatic cancer cells, significantly inhibiting their growth by inducing G2/M cell cycle arrest and apoptosis. In vivo studies using Capan-2 and CFPAC-1 xenograft models showed that HzMUC1-MMAE markedly reduced tumor growth without notable toxicity [132]. At last, MUC1 enhanced the expression of multidrug resistance (MDR) genes, such as ABCC1, leading to higher levels of the MRP1 protein, which is involved in drug efflux. MUC1 also interacted with hypoxia-inducible factor 1-alpha (HIF-1a), promoting glycolysis and glutamine metabolism, further contributing to chemoresistance. Targeted therapies against MUC1, including vaccines, antibodies, and inhibitors, have shown promise in improving chemotherapy efficacy and patient survival. The study highlighted the need for further research into MUC1′s role in maintaining cancer stem cell characteristics and its detailed mechanisms of action to develop more effective treatments for PC [133].

## 3. Conclusions

In summary, the present review emphasizes the roles of some microRNAs, GATA6, L1CAM, and MUC1 in pancreatic cancer (Table 3). L1CAM facilitates tumor growth by facilitating cell multiplication, invasion, and resistance to chemotherapy while also affecting the dynamics of the tumor environment. On the other hand, abnormal MUC1 expression is linked to tumor resistance to chemotherapy and a grim prognosis in pancreatic cancer. Moreover, levels of microRNA and GATA6 expression could act as indicators for foreseeing how patients respond to chemotherapy and their survival rates in types of PDAC. These discoveries provide insights into the processes implicated in PDAC progression, hinting at the benefits of tailored treatments that target these pathways for enhancing treatment results.

## Figures and Tables

**Figure 1 cimb-47-00347-f001:**
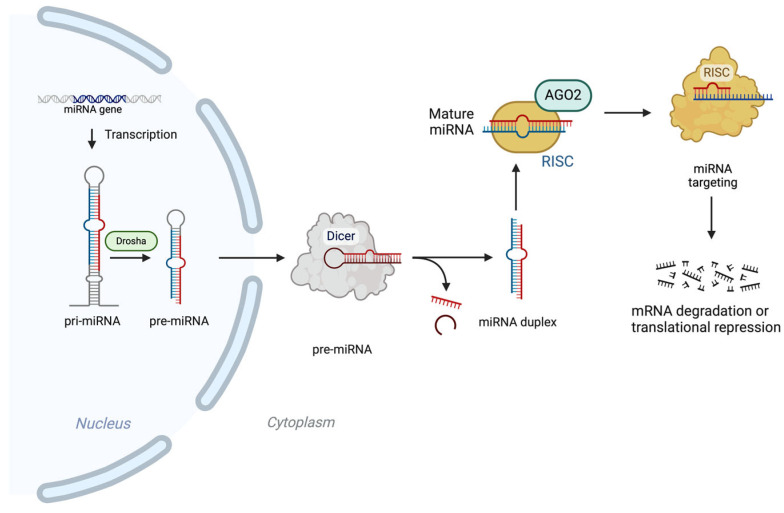
MicroRNAs’ (miRNA) synthesis and way of action. miRNAs are produced by RNA polymerase II, and miRNAs undergo processing by the Drosha complex to become precursor miRNAs (pre-miRNAs) and are further modified by Dicer in the cytoplasm to form miRNA duplexes. When the miRNA duplex is processed by Dicer, one of the strands is loaded onto the RNA-induced silencing complex (RISC), and the other is removed. The guide strand is retained to direct RISC to the target mRNA through complementary base pairing that occurs mostly at the 3′ untranslated region (UTR). If the miRNA has near-perfect sequence similarity to the target mRNA, AGO proteins in the RISC affect the endonucleolytic cleavage of the mRNA, resulting in its degradation. The cleaved mRNA is then further broken down by exonucleases, thus suppressing gene expression [34,35]. Created with BioRender.com; https://BioRender.com/j34i195 (accessed on 8 February 2025).

**Figure 2 cimb-47-00347-f002:**
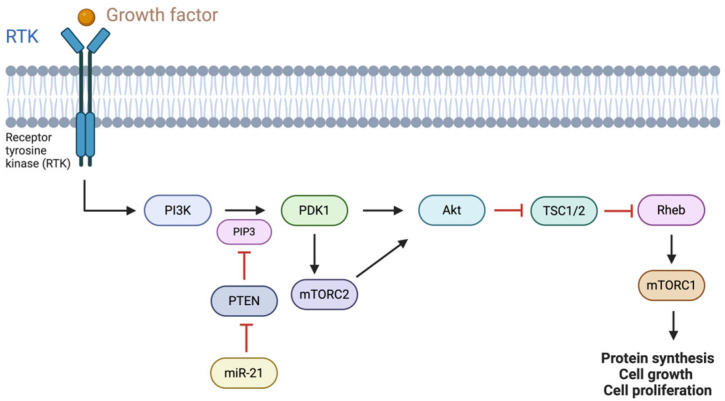
The AKT signaling pathway is known to be regulated by miR-21 through the targeting and suppression of the tumor suppressor phosphatase and tensin homolog (PTEN). Normally, PTEN dephosphorylates phosphatidylinositol-3,4,5-triphosphate (PIP3), preventing AKT activation. However, reduced PTEN levels due to miR-21 result in increased PIP3 levels and, thus, sustained phosphorylation and the activation of AKT. This boosts cell survival, proliferation, and apoptosis resistance, all leading to tumor progression. The figure shows how miR-21 affects the PTEN/AKT pathway and its oncogenic role in cancer development [71,72]. Created with BioRender.com; https://BioRender.com/f76q527 (accessed on 8 February 2025).

**Figure 3 cimb-47-00347-f003:**
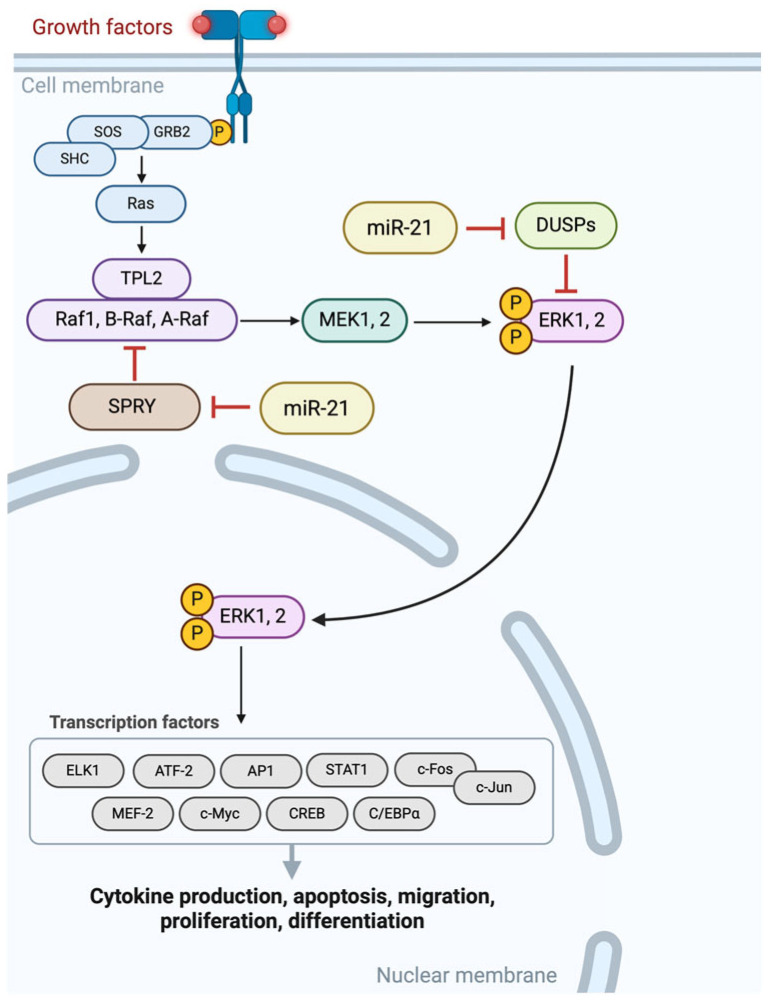
MiR-21 modulates the MAPK/MEK/ERK signaling pathway directly by downregulating the SPRY and DUSPs that normally suppress ERK activation. MiR-21 facilitates sustained MEK ERK phosphorylation by suppressing certain inhibitors, thereby enhancing cell proliferation and survival. Dysregulation fuels unchecked growth in tumor cells, fostering resistance in apoptosis that occurs sporadically amidst rapid proliferation. The figure illustrates the role of miR-21 in modulating the MAPK/MEK/ERK pathway, emphasizing its deep oncogenic influence in cancer development processes [73,74,75,76]. Created with BioRender.com; https://BioRender.com/e16k991 (accessed on 8 February 2025).

**Table 2 cimb-47-00347-t002:** The table lists the potentially detected pancreatic cancer miRNA biomarkers in various biological fluids including serum, liquid biopsy, urine, pancreatic juice, pancreatic cyst fluid, saliva, bile, and feces. These biomarkers may help in noninvasive cancer detection, prognosis, and monitoring by measuring their levels in different tissue and fluid samples.

Tissue/Fluid Source	Potential miRNA Biomarker
Serum	miR-21 [4,40,41,42,43], miR-210-3p [43], miR-210 [4,44], miR-196a [4,45], miR-451a [46], miR-155 [4,47], miR-1469 [37], miR-125a-3p [37], miR-642b-3p [48], miR-34a [40], miR-1290 [49,50], miR-1246 [50]
Liquid biopsy	miR-10b [51], miR-21 [52], miR-181a [51], miR-1246 [53]
Urinary biomarkers	miR-143 [54], miR-1246 [55]
Pancreatic juice biomarkers	miR-21 [56], miR-155 [56]
Pancreatic cyst fluid	miR-21 [57], miR-221 [57], miR-142-3p [58,59]
Salivary fluid	miR-23a [60], miR-21 [40,60,61], miR-1246 [62], miR-34a [40], miR-155 [40], miR-200b [40], miR-196a [63]
Biliary fluid	miR-10b [47,64], miR-155 [47,64], miR-212 [47,64], miR-200a [65], miR-200b [65]
Feces	miR-181b [66], miR-210 [66], miR-155 [67], miR-196a [66,67], miR143 [67]

**Table 3 cimb-47-00347-t003:** This is a table that summarizes the papers used in the review of miRNAs, GATA, L1CAM, and MUC1, concerning the category that distinguishes nonclinical studies and clinical studies.

	Clinical Study	Nonclinical Study
miRNAs	[4,15,31,37,40,41,42,43,44,45,46,47,48,49,50,51,52,53,54,55,56,57,58,59,60,61,62,63,65,66,67,68,69,76,82]	[7,8,9,10,13,14,16,17,18,29,30,34,35,36,38,39,64,70,71,72,73,74,75,77,78,79,80,81,83,84,85]
GATA	[91,92,95,96,98,99,101]	[20,21,86,87,88,89,90,93,94,97,100,102,103,104,105,106]
L1CAM	[23,108,110,118]	[22,24,107,109,111,112,113,114,115,116,117,119]
MUC1	[122,128]	[25,26,27,28,120,121,123,124,125,126,127,129,130,131,132,133]

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
