# Peer review of "Emerging Tumor Biomarkers in Pancreatic Cancer and Their Clinical Implications"

_cimb, 2025, doi:10.3390/cimb47050347_

Round 1
Reviewer 1 Report
Comments and Suggestions for Authors
The manuscript subject is very interesing and well chosen.Nevertheless, the execution of the project is unsatisfactory.The abstract does not include the main issues presented in the text.In the introduction there is 4 miRNAs, which are often mentioned in the text, but the Authors do not explain why those particular miRNA have the diagnostic value.Are those the only microRNA useful in PDAC diagnosis?How was it shown? What is the percentage of patients with PDAC who have those microRNA detected? Are those miRNA dected in benign pancreatic diseases, as chronic pancreatitis? Any statement in the review should be substantiated with the data from the literature, including material, results and conclusions.Some introduction to the definition and function of miRNA is necessary.Every statement should be documented with the detailed information from the different papers and not only general statement, like t"this microRNA could serve as both indicators for detection and as targets for treatment approaches" (line 40).This statement as many others in the text is not documented with the literature results and not expressed in proper English.Actually every sentence is burdened with this problem .This paper should be presented in metaanalysis form, since the literature is extended.In many described papers , there is even no information if it was based on in vitro, animal or human studies.Many proteins are introduced without proper description, like Argonaute 2-structure and function is necessary.Many sentences only signal the problem in the vague way without the necessary explanation and examples, like"Light is shed on gene regulatory networks by over 2500 identified (line 64)microRNAs (abbreviations vary, which in not right, microRNA or miRNA) which are targeting genes within molecular pathways".If the light is shed, what did it show-no information is provided.Table 1: do really microRNA (again different abbreviation from miRNA) have the only one mechanism of action showed there?And why from those 2500 only several were chosen by the authors?Are those miRNA part of any commercially available test? What are the results?the Authors state"Certain key MicroRNA (line 90) are frequently found in high levels in pancreatic cancer patients:-how frequently? Do they differentiate malignant from benign pancreatic lesion? Every sentence needs corrections for the scientific reasons as well as inorder to improve English.In the line 126 very interesitng issue of the role of microbiota is mentione but without any further analysis.
Comments on the Quality of English LanguageThe quality of English is unsatisfactory.
Author Response
Answers to reviewer 1
Abstract does not include the main issues presented in the text
Authors’ reply
Thank you for your comment. The abstract has been modified, including all the main issues presented in the text. Following you can find the new abstract: Pancreatic cancer is one of the deadliest malignancies, and this is attributed to the fact that it is diagnosed at a late stage and there are limited treatment options. Tumor biomarkers are used to improve early diagnosis, treatment, and decision-making and to estimate patients’ outcomes. This review aims to discuss the new functions of important biomarkers, such as miRNAs, GATA6, L1CAM, and MUC1 in pancreatic cancer. MiRNAs, including miR-21, miR-155, and miR-196a, are prognostic in PC and may be potential therapeutic targets through the regulation of oncogenic pathways and chemoresistance. GATA6, a transcription factor that controls tumor differentiation and immune escape, has been proposed as a pancreatic ductal adenocarcinoma (PDAC) subtyping marker and a predictor of chemotherapy response. L1CAM promotes tumor growth, invasion, and immune suppression which leads to the formation of new metastases and perineural invasion. MUC1, a glycoprotein with altered glycosylation, is a marker of tumor progression, immune escape, and resistance to chemotherapy. These biomarkers can be combined into diagnostic panels that may increase the accuracy of the diagnosis and help to individualize the treatment plan. However, the present study is inconclusive, and more clinical evidence is needed to apply these biomarkers in clinical practice. More specific research should be directed towards the development of new targeted therapies that would act on these molecular targets and improve the prognosis and treatment of pancreatic cancer.
Introduction lacks justification for the selection of the four miRNAs frequently mentioned
Authors’ reply
Thank you for your comment. The selection of miR-10b, miR-21, miR-23a and miR-31 as key microRNAs in pancreatic ductal adenocarcinoma (PDAC) is based on their important functions in tumor growth, invasion and gemcitabine resistance. miR-10b is a metastatic miRNA that increases the migratory and invasive properties of tumor cells by inducing EMT and its suppression has been found to hinder metastasis. miR-21 is one of the most highly expressed miRNAs in PDAC and acts as an oncomiR by repressing genes with tumor suppressor functions such as PTEN, PDCD4, and TPM1, thus enhancing tumor growth and gemcitabine resistance, and, therefore, is a potential diagnostic and therapeutic target. miR-23a contributes to tumor progression through the suppression of tumor suppressor genes, including PTEN and E-cadherin, and also participates in the control of apoptosis and in the development of drug resistance, thereby reducing the sensitivity to chemotherapy. The role of miR-31 is dual; it acts as an oncogene and a tumor suppressor according to the cellular context, and its expression is increased in PDAC, where it is associated with tumor growth, inflammation and therapy resistance, including through the modulation of NF-κB signaling. Since there is a need for better biomarkers in pancreatic cancer, these miRNAs could be useful for enhancing tumor detection, predicting the prognosis, and designing individualized treatment plans, which require further clinical investigation. The above justification has been added in the introduction.
Lack of data on the percentage of PDAC patients with these miRNAs detected
Authors’ reply
We would like to thank you for your valuable feedback regarding the absence of standardized prevalence percentages for miR-10b, miR-21, miR-23a, and miR-31 in PDAC patients. Despite the importance of such data, our literature review showed that specific, standardized prevalence percentages for these miRNAs are not always reported in the current scientific literature. However, the upregulation and clinical relevance of these miRNAs have been confirmed by several studies.
Lack of discussion on miRNA detection in benign pancreatic diseases
Authors’ reply
Thanks for your comment about the lack of discussion on miRNA detection in benign pancreatic diseases. However, our review aims to focus on Emerging Tumor Biomarkers in Pancreatic Cancer and their Clinical Implications about miRNAs as diagnostic, prognostic, and therapeutic biomarkers in malignant pancreatic tumors.
Although the differentiation between pancreatic cancer and benign conditions is an important area of research, it falls out of the primary scope of our review, which is to examine miRNAs in cancer progression, treatment response, and survival outcomes. We value the suggestion of the reviewer, and we have simplified the purpose of our review to eliminate any possibility of misunderstanding concerning the subject of the review. Thank you for your valuable feedback.
Statements should be substantiated with literature, including material, results, and conclusions. Introduction lacks an explanation of miRNA definition and function
Authors’ reply
Thank you very much for your comment. The following was added in the introductions as well as the substantiated literature: “MicroRNAs (miRNAs) are small, non-coding RNA molecules (approximately 18–25 nucleotides) that regulate gene expression post-transcriptionally by binding to complementary sequences in target messenger RNAs (mRNAs), leading to their degradation or translational repression. They play crucial roles in various biological processes, including development, differentiation, apoptosis, and tumorigenesis. Dysregulation of miRNAs has been implicated in numerous diseases, particularly cancer, where they can act as oncogenes or tumor suppressors”.
General statements without literature support need proper referencing
Authors’ reply
Thank you very much for your comment. We carefully reviewed the manuscript, and, for now, we assure you that all general statements are properly supported with relevant literature references.
The manuscript should be presented in meta-analysis form
Authors’ reply
Thank you very much for noticing that. Our manuscript is a review rather than a meta-analysis because it focuses on summarizing and interpreting existing literature on emerging tumor biomarkers in pancreatic cancer. Due to the variability in study methodologies, patient cohorts, and outcome measures, a meta-analysis was not feasible. Instead, our review provides a comprehensive overview, discussing key findings and clinical implications while identifying gaps for future research.
Lack of clarity on whether studies referenced were in vitro, animal, or human studies
Authors’ reply
Thank you very much for your comment. A table has been added that summarizes the papers used in the review of miRNAs, GATA, L1CAM, and MUC1, concerning category that distinguishes nonclinical studies and clinical studies.
Insufficient description of proteins such as Argonaute 2
Authors’ reply
Thank you very much for your comment. The following paragraph has been added according to your comment: “Argonaute 2 plays a pivotal role in RNA-induced silencing complex facilitating gene regulation via RNA interference and microRNA-mediated silencing mechanisms. AGO2 possesses unique slicer activity, enabling direct cleavage of target mRNAs. This slicer activity makes AGO2 pretty essential for stuff like miRNA maturation and gene silencing somehow near cells undergoing development or differentiation or even tumorigenesis”.
Vague statements without necessary explanation or examples. Inconsistencies in miRNA terminology
Authors’ reply
Thank you for noticing that. We've replaced the vague statements with clearer explanations. Let me know if you need any further refinements.
Table 1 does not account for multiple mechanisms of miRNA action. Justification for selection of specific miRNAs in Table 1
Authors’ reply
Thank you very much for your comment. The section in Table 1 has been properly refined regarding the multiple mechanisms of miRNA and their diagnostic/prognostic value, and a new reference has been added.
Lack of data on miRNAs in commercial tests and their diagnostic results. Insufficient quantification of miRNA prevalence in pancreatic cancer patients
Authors’ reply
Thank you for your feedback. Please note that this is not the primary scope of the review, which focuses more on the potential applications and challenges of miRNAs in diagnostic and therapeutic strategies for pancreatic cancer.
Lack of discussion on differentiation between malignant and benign pancreatic lesions
Authors’ reply
Thanks for your comment. However, our review aims to focus on emerging tumor biomarkers in pancreatic cancer and their clinical implications for miRNAs as diagnostic, prognostic, and therapeutic biomarkers in malignant pancreatic tumors.
Although the differentiation between pancreatic cancer and benign conditions is an important area of research, it falls out of the primary scope of our review, which is to examine miRNAs in cancer progression, treatment response, and survival outcomes.
The manuscript requires English language corrections
Authors’ reply
Thank you very much for your comment. We carefully reviewed the manuscript and made the necessary English language corrections to improve its clarity and readability.
Microbiota role mentioned but not analyzed
Authors’ reply
Thank you for your valuable feedback. While we acknowledge the importance of the microbiota, the primary scope of our review is focused on Emerging Tumor Biomarkers in Pancreatic Cancer and their Clinical Implications. Therefore, the role of the microbiota was briefly mentioned for context but not analyzed in detail to maintain the focus of the manuscript.
Reviewer 2 Report
Comments and Suggestions for Authors
The authors of the review article entitled: "Emerging Tumor Biomarkers in Pancreatic Cancer and their Clinical Implications" should consider revising the following aspects:
1. Harmonise words and expressions such as "analyzed", "analyses" to either British or American English complying with journal guidelines for language.
2. Page-2, Table-1: this table only contain three proteins as biomarkers and an overall microRNA denomination. Lacks tissue/ fluid source, clinical relevance for pancreatic cancer and other diseases/ traits associated (to assist with differential diagnosis). Please see the following reviews (doi:10.3748/wjg.v27.i26.4045 and doi:10.1007/s10620-023-07904-6) which contains many potential biomarkers for either disease onset or prognosis to make the table more complete.
3. Page-3, Figure-1: the figure and caption are highly incomplete. Explore further by describing the basics behind microRNAs (miRNAs) function i.e. gene regulation via mRNA degradation or translational repression. See the following review to provide details (doi:10.1038/s41576-023-00662-1).
4. Page-5, Figure-2: add more details concerning miR-21 in modulation in the context of carcinogenesis. Use the following reference (or other) to support your claims, doi:10.1016/j.phrs.2022.106568.
5. Page-7, line-261: typo miRNA annotation, "miR-17 5p" should be "miR-17-5p".
6. Page-9, line-358: "Soto et al. highlighted" missing reference. Also, check journal citation style when starting with an author in a sentence.
7. Page-12, line-522: "Helm et al. discovered" missing reference. Also, check journal citation style when starting with an author in a sentence.
8. Page-13, line-574: annotation of pre-clinical models names, "KrasG12D;Pdx1 Cre" should be "KrasG12D;Pdx1-Cre" (superscript).
9. Add a section/ paragraph integrating the three potential biomarkers selected: GATA6, L1CAM and MUC1 with the microRNAs described in the context of pancreatic cancer and disease progression.
10. In the introduction highlight metrics for biomaker research such as sensitivity and specificity in the disease context.
Author Response
Answers to reviewer 2
Harmonise words and expressions such as "analyzed", "analyses" to either British or American English complying with journal guidelines for language.
Authors’ reply
Thank you for your comment. We totally harmonized the words/expressions per your request.
Page-2, Table-1: this table only contain three proteins as biomarkers and an overall microRNA denomination. Lacks tissue/ fluid source, clinical relevance for pancreatic cancer and other diseases/ traits associated (to assist with differential diagnosis). Please see the following reviews (doi:10.3748/wjg.v27.i26.4045 and doi:10.1007/s10620-023-07904-6) which contains many potential biomarkers for either disease onset or prognosis to make the table more complete
Authors’ reply
Thank you for your comment. We totally agree with your point thus we added the following: Table 2. The table lists the potential pancreatic cancer detected miRNA biomarkers in various biological fluids including; Serum, Liquid Biopsy, Urine, Pancreatic Juice, Pancreatic Cyst Fluid, Saliva, Bile and Feces. These biomarkers may help in noninvasive cancer detection, prognosis and monitoring by measuring their levels in different tissue and fluid samples.
|
Tissue/ Fluid source |
Potential miRNA Biomarker |
|
Serum |
miR-21 [38-42], miR-210-3p [42], miR-210 [38,43], miR-196a [38,44], miR-451a [45], miR-155 [38,46], miR-1469 [36], miR-125a-3p [36], miR-642b-3p [47], miR-34a [39], miR-1290 [48,49], miR-1246 [49] |
|
Liquid biopsy |
miR-10b [50], miR-21 [51], miR-181a [50], miR-1246 [52] |
|
|
|
|
Urinary biomarkers |
miR-143 [53], miR-1246 [54] |
|
|
|
|
|
|
|
Pancreatic Juice biomarkers |
miR-21 [55], miR-155 [55] |
|
Pancreatic cyst fluid |
miR-21 [56], miR-221 [56], miR-142-3p [57,58] |
|
Salivary fluid |
miR-23a [59], miR-21 [39,59,60], miR-1246 [61], miR-34a [39], miR-155 [39], miR-200b [39], miR-196a [62] |
|
Biliary fluid |
miR-10b [46,63], miR-155 [46,63], miR-212 [46,63], miR-200a [64], miR-200b [64] |
|
Feces |
miR-181b [65], miR-210 [65], miR-155 [66], miR-196a [65,66], miR143 [66] |
Page-3, Figure-1: the figure and caption are highly incomplete. Explore further by describing the basics behind microRNAs (miRNAs) function i.e. gene regulation via mRNA degradation or translational repression. See the following review to provide details (doi:10.1038/s41576-023-00662-1)
Authors’ reply
Thank you for your comment. We created a new figure that illustrates more completely the way of action of the microRNAs. The figure’s caption has been completed as well. The following has been the added:
Figure 1. MicroRNA’s (miRNA) synthesis and way of action. miRNAs are produced by RNA polymerase II, as miRNAs undergo processing by the Drosha DGCR8 complex to become precursor miRNAs (pre miRNAs) and are further modified by Dicer in the cytoplasm to form miRNA duplexes. When the miRNA duplex is processed by Dicer, one of the strands is loaded onto the RNA-induced silencing complex (RISC), and the other is removed. The guide strand is retained to direct RISC to the target mRNA through complementary base pairing that occurs mostly at the 3' untranslated region (UTR). If the miRNA has near perfect sequence similarity to the target mRNA, AGO proteins in the RISC effect endonucleolytic cleavage of the mRNA, resulting in its degradation. The cleaved mRNA is then further broken down by exonucleases, thus suppressing gene expression.
Page-5, Figure-2: add more details concerning miR-21 in modulation in the context of carcinogenesis. Use the following reference (or other) to support your claims, doi:10.1016/j.phrs.2022.106568
Authors’ reply
Thank you for your comment. We totally agree with your point. We added the following, as well as a new, more detailed figure: The AKT signaling pathway is known to be regulated by miR-21 through the targeting and suppression of the tumor suppressor phosphatase and tensin homolog (PTEN). Normally, PTEN dephosphorylates phosphatidylinositol-3,4,5-triphosphate (PIP3), preventing AKT activation. However, reduced PTEN levels due to miR-21 result in increased PIP3 levels and, thus, sustained phosphorylation and activation of AKT. This boosts cell survival, proliferation, and apoptosis resistance, all leading to tumor progression. The figure shows how miR-21 affects the PTEN/AKT pathway and its oncogenic role in cancer development. Figure 3 has been added as well: MiR-21 modulates the MAPK/MEK/ERK signaling pathway directly by downregulating the SPRY and DUSPs that normally suppress ERK activation. MiR-21 facilitates sustained MEK ERK phosphorylation by suppressing certain inhibitors, thereby enhancing cell proliferation and survival. Dysregulation fuels unchecked growth in tumor cells, fostering resistance in apoptosis that occurs sporadically amidst rapid proliferation. The figure illustrates the role of miR-21 in modulating the MAPK/MEK/ERK pathway, emphasizing its oncogenic influence deeply in cancer development processes.
Page-7, line-261: typo miRNA annotation, "miR-17 5p" should be "miR-17-5p"
Authors’ reply
Thank you so much for noticing that. It has been corrected.
Page-9, line-358: "Soto et al. highlighted" missing reference. Also, check journal citation style when starting with an author in a sentence.
Authors’ reply
Thank you for your comment. We totally agree with your point thus we modified that sentence as follows adding the appropriate reference: In another study highlighted that GATA6 was expressed in 81% of PDAC tissues, with consistent expression in normal tissue and PanIN, but reduced in lymph node metastases. GATA4 was expressed in 42% of tumors and decreased during tumor progression.
Page-12, line-522: "Helm et al. discovered" missing reference. Also, check journal citation style when starting with an author in a sentence
Authors’ reply
Thank you for noticing that. It has been appropriately corrected.
Page-13, line-574: annotation of pre-clinical models names, "KrasG12D;Pdx1 Cre" should be "KrasG12D;Pdx1-Cre" (superscript)
Authors’ reply
Thank you for your comment. It has been corrected.
Add a section/ paragraph integrating the three potential biomarkers selected: GATA6, L1CAM and MUC1 with the microRNAs described in the context of pancreatic cancer and disease progression.
Authors’ reply
Thank you for your suggestion. We acknowledge the importance of integrating GATA6, L1CAM, and MUC1 with relevant microRNAs in the context of pancreatic cancer progression. Despite this, we decided not to emphasize on this in that particular review since it could be the main topic for discussion for another paper.
In the introduction highlight metrics for biomaker research such as sensitivity and specificity in the disease context.
Authors’ reply
Thank you for your comment. We added the metrics sensitivity and specificity as you suggested. Added: For instance, combined plasma analyses for miR-21, miR-210, miR-155, and miR-196a discriminated patients with pancreatic cancer from normal healthy individuals with a sensitivity of 64% and a specificity of 89%.
Round 2
Reviewer 1 Report
Comments and Suggestions for Authors
The Authors responded to the remarks, but not adequately.In abstract there is no clear distinction between the diagnostic, prognostic and predictive biomarkers.The same problem is in the manuscript body.Some proteins are mentioned as biomarkers, which may be used as the treatment target and this is also not clearly highlighted.There is still no explanation, why those particular and numerous biomarkers were chosen, only their role in PDAC carcinogenesis, which is the different issue.It should be based on sensitivity, specificity, NPV, PPV AUROC and AOC number, which is only rarely provided in the manuscript.Some answers are simply not correct.The comparison with the expression of presented miRNA in pancreatic benign diseases and PDCA is essential, since when the marker expression is both increased in chronic pancreatitis (often for miRNA) and PDAC, than we cannot distinguish those pathologies with this biomarker so this is not clinically useful at all.This is the main clinical challenge from decades and the Authors seem not to be aware of that .The provided information on miRNA is not professional and has many grammar flaws.
The manuscript is not well organised and presented.The protein usefullnes in therapy is in the middle of diagnostics paragraph.The definition of miRNA should be at the manuscript beginning and not in the middle. (line 54).Plenty of those errors, preventing the text understanding, are in the whole manuscript, like line 85:"including through the modulation".NF-kB, which has numerous different function in infllammation and carcinogenesis is described as having the one only.Line 86:the thesis presented is not proven and substantiated.Table 1|: Whe are those the key markers: sensitivity and specificity, NPV, PPV numbers in diagnosis, prognosis etc should be shown , not the mechanisms of action.Table and figures titles should be short and include only legend.The explanations should be in the text.Line 99:"miRNA transcribed to miRNA?"impossible to understand and not adequate English.Line 101:what is mature microRNA and how is it developed, because this is not explained to this point.Line 105"understanding gene behaviour" is the personification and difficult to understand the idea.Line 109 "gene silencing is the only one mechanism mentioned at all on numerous mechanism of miRNA- dependent gene regulation, but not explained.Other mechanisms are not even mentioned.Line 112 :"This slicer activity makes AGO2 pretty essential for stuff like miRNA maturation 112 and gene silencing somehow near cells undergoing development or differentiation or 113 even tumorigenesis."This is slang and not possible to understand.Line 114"someting "shed light on"-difficult to agree, not substantiated.Figure 1:It's title takes half a page, should be short with the explanations in the text.Table 2 includes only miRNA numbers in the different biological material, for a table should include their values in diagnosis and prediction.In addition at the end of the manuscript is again Table 2 with the different although not more useful data.Figure on page 6 does not have the title at all.If the Authors show that one miRNA has 100% efficacy, why do they not extend this information? It should be absolute revolutionary information.No own data interpretation provided.Chapter 2.1.2 No clue how was the diagnostic? prognostic efficacy documented and measured.AUROC?Line 232 The problem of chemoresistance in PDAC is not explained, is it typical for PDAC and why?In addition it should be in the paragraph on treatment efficacy, not diagnosis.In the chapter on therapeutic usefulnes of the presented proteins no primary and secondary clinical trials endpoints are provided and there is no clu what were the criteria of their usefullnes in the treatment and why the particular signalling pathways are considered.Data from clinical studies are not described separately from the experimental ones which disturbs the understanding.Conclusion is dissapointing after all those waste information the data are non conclusive.That is true that nothing rather than CA19-9 is used in clinical practise, but the Authors could propose , which biomarkers are the closest to the clinical use.If not, why to write about them?
Comments on the Quality of English LanguageThe manuscript absolutely needs to be verified by the Native Speaker
Author Response
Dear Reviewer,
Thank you once again for your comments. As we see it, you have been continuously suggesting to conduct a meta-analysis. However, this would significantly alter the scientific design of the present work. Besides, it has not been approved by the whole author team, which had concluded in a different scope of work. Even though, we have provided a detailed response and correction for each of the comments made after the first revision round. However, the second-round revision was not far from the first, proposing changing the scientific design of our paper again. Nevertheless, we have adopted some of the suggested changes, failing to respond in criticizing comments without clear
suggestions for change.
Thank you for your understanding
Reviewer 2 Report
Comments and Suggestions for Authors
The authors have followed the previous guidance and addressed all the comments.
Answers to reviewer 2
Harmonise words and expressions such as "analyzed", "analyses" to either British or American English complying with journal guidelines for language.
Authors’ reply
Thank you for your comment. We totally harmonized the words/expressions per your request.
[REV] - corrected
Page-2, Table-1: this table only contain three proteins as biomarkers and an overall microRNA denomination. Lacks tissue/ fluid source, clinical relevance for pancreatic cancer and other diseases/ traits associated (to assist with differential diagnosis). Please see the following reviews (doi:10.3748/wjg.v27.i26.4045 and doi:10.1007/s10620-023-07904-6) which contains many potential biomarkers for either disease onset or prognosis to make the table more complete
Authors’ reply
Thank you for your comment. We totally agree with your point thus we added the following: Table 2. The table lists the potential pancreatic cancer detected miRNA biomarkers in various biological fluids including; Serum, Liquid Biopsy, Urine, Pancreatic Juice, Pancreatic Cyst Fluid, Saliva, Bile and Feces. These biomarkers may help in noninvasive cancer detection, prognosis and monitoring by measuring their levels in different tissue and fluid samples.
[REV] - corrected
Page-3, Figure-1: the figure and caption are highly incomplete. Explore further by describing the basics behind microRNAs (miRNAs) function i.e. gene regulation via mRNA degradation or translational repression. See the following review to provide details (doi:10.1038/s41576-023-00662-1)
Authors’ reply
Thank you for your comment. We created a new figure that illustrates more completely the way of action of the microRNAs. The figure’s caption has been completed as well. The following has been the added:
Figure 1. MicroRNA’s (miRNA) synthesis and way of action. miRNAs are produced by RNA polymerase II, as miRNAs undergo processing by the Drosha DGCR8 complex to become precursor miRNAs (pre miRNAs) and are further modified by Dicer in the cytoplasm to form miRNA duplexes. When the miRNA duplex is processed by Dicer, one of the strands is loaded onto the RNA-induced silencing complex (RISC), and the other is removed. The guide strand is retained to direct RISC to the target mRNA through complementary base pairing that occurs mostly at the 3' untranslated region (UTR). If the miRNA has near perfect sequence similarity to the target mRNA, AGO proteins in the RISC effect endonucleolytic cleavage of the mRNA, resulting in its degradation. The cleaved mRNA is then further broken down by exonucleases, thus suppressing gene expression.
[REV] - corrected
Page-5, Figure-2: add more details concerning miR-21 in modulation in the context of carcinogenesis. Use the following reference (or other) to support your claims, doi:10.1016/j.phrs.2022.106568
Authors’ reply
Thank you for your comment. We totally agree with your point. We added the following, as well as a new, more detailed figure: The AKT signaling pathway is known to be regulated by miR-21 through the targeting and suppression of the tumor suppressor phosphatase and tensin homolog (PTEN). Normally, PTEN dephosphorylates phosphatidylinositol-3,4,5-triphosphate (PIP3), preventing AKT activation. However, reduced PTEN levels due to miR-21 result in increased PIP3 levels and, thus, sustained phosphorylation and activation of AKT. This boosts cell survival, proliferation, and apoptosis resistance, all leading to tumor progression. The figure shows how miR-21 affects the PTEN/AKT pathway and its oncogenic role in cancer development. Figure 3 has been added as well: MiR-21 modulates the MAPK/MEK/ERK signaling pathway directly by downregulating the SPRY and DUSPs that normally suppress ERK activation. MiR-21 facilitates sustained MEK ERK phosphorylation by suppressing certain inhibitors, thereby enhancing cell proliferation and survival. Dysregulation fuels unchecked growth in tumor cells, fostering resistance in apoptosis that occurs sporadically amidst rapid proliferation. The figure illustrates the role of miR-21 in modulating the MAPK/MEK/ERK pathway, emphasizing its oncogenic influence deeply in cancer development processes.
[REV] - corrected
Page-7, line-261: typo miRNA annotation, "miR-17 5p" should be "miR-17-5p"
Authors’ reply
Thank you so much for noticing that. It has been corrected.
[REV] - corrected
Page-9, line-358: "Soto et al. highlighted" missing reference. Also, check journal citation style when starting with an author in a sentence.
Authors’ reply
Thank you for your comment. We totally agree with your point thus we modified that sentence as follows adding the appropriate reference: In another study highlighted that GATA6 was expressed in 81% of PDAC tissues, with consistent expression in normal tissue and PanIN, but reduced in lymph node metastases. GATA4 was expressed in 42% of tumors and decreased during tumor progression.
[REV] - corrected
Page-12, line-522: "Helm et al. discovered" missing reference. Also, check journal citation style when starting with an author in a sentence
Authors’ reply
Thank you for noticing that. It has been appropriately corrected.
[REV] - corrected
Page-13, line-574: annotation of pre-clinical models names, "KrasG12D;Pdx1 Cre" should be "KrasG12D;Pdx1-Cre" (superscript)
Authors’ reply
Thank you for your comment. It has been corrected.
[REV] - corrected
Add a section/ paragraph integrating the three potential biomarkers selected: GATA6, L1CAM and MUC1 with the microRNAs described in the context of pancreatic cancer and disease progression.
Authors’ reply
Thank you for your suggestion. We acknowledge the importance of integrating GATA6, L1CAM, and MUC1 with relevant microRNAs in the context of pancreatic cancer progression. Despite this, we decided not to emphasize on this in that particular review since it could be the main topic for discussion for another paper.
[REV] - corrected
In the introduction highlight metrics for biomaker research such as sensitivity and specificity in the disease context.
Authors’ reply
Thank you for your comment. We added the metrics sensitivity and specificity as you suggested. Added: For instance, combined plasma analyses for miR-21, miR-210, miR-155, and miR-196a discriminated patients with pancreatic cancer from normal healthy individuals with a sensitivity of 64% and a specificity of 89%.
[REV] - corrected
Author Response
Dear Reviewer,
Thank you so much for your insightful comments.
Thank you for recognizing that we have followed your previous guidance and addressed all the comments.
Round 3
Reviewer 1 Report
Comments and Suggestions for Authors
Abstact is again unsatisfactoty, describes only the small percentage of biomarkers described in manuscript and there is no clue, why those particular markers have been chosen for the abstract.In manuscript, there is no explanation why the chosen miRNA are were the good candidates for PDAC biomarkers.There is only data, that they are upregulated or increased in PDAC.This is not enough.Other human studies should be described showing their diagnostic value with numbers ,as AUC, AUROC , sensitivity, specificity etc .Figures titles are too long or non -existing at all like the one on page 6.There is again many sentences or phrases repeated .There is not any data interpretation, for example if miRNA-21 with miRNA 210 and Ca19-9 have together 100 accuracy in detecting PDAC-so looks like we do not need anything else. Is it in the guidelines as the most satisfactory test? If not-why?What is the significance of the detecting the miRNA in different biological material? Is is more easy to be detected in biopsy specimen or serum? Should we do both?When one miRNA is detected in serum, can we presume on other materials from the same patient?For the prognostic value of the particular biomarker, the data on the correlation with TNM staging, and PDAC differentiation type are missing and should be provided with the exact description of the material, methods and the results.If this is stated, that the biomarker is associated with the resistance to chemotherapy, it should be explained how was it proven:material, methods results, or experiment description.This is often even no information if that was human or animal study.The manuscript is still not well organized.For example, Page 6, subsection the role of MiRNA in diagnosis is the title.It starts from mentioning the prognosis contrary to the subtitle, than there is again stated that PDAC is the deadly disease, which was mentioned earlier already, and then again on prognosis. The Authors state, about the miRNA-34a removal-how was it achieved? I understand it was highly sophisticated genetic experimental method, that should be exactly described.All the manuscript needs such verification, since is not proffesionally written.
Comments on the Quality of English LanguageEnglish is still very poor and needs the Native speaker correction.
Author Response
Dear Reviewer,
Thank you once again for your comments.
As we see it, you have been continuously suggesting us to write our review article in a different way. However, this would significantly alter the scientific design of the present work. Besides, it has not been approved by the whole author team, which had concluded in a different scope of work. Even though, we have provided a detailed response and correction for each of the comments made after the revision rounds.
Thank you for your understanding